# Targeting NF-κB by the Cell-Permeable NEMO-Binding Domain Peptide Improves Albuminuria and Renal Lesions in an Experimental Model of Type 2 Diabetic Nephropathy

**DOI:** 10.3390/ijms21124225

**Published:** 2020-06-13

**Authors:** Lucas Opazo-Ríos, Anita Plaza, Yenniffer Sánchez Matus, Susana Bernal, Laura Lopez-Sanz, Luna Jimenez-Castilla, Daniel Carpio, Alejandra Droguett, Sergio Mezzano, Jesús Egido, Carmen Gomez-Guerrero

**Affiliations:** 1Division of Nephrology, School of Medicine, Universidad Austral de Chile, 5090000 Valdivia, Chile; lucasopazo78@gmail.com (L.O.-R.); anitaplazaflores@yahoo.com (A.P.) yenniffersanchezmatus@gmail.com (Y.S.M.); dcarpiop@gmail.com (D.C.); m.aledroguett@gmail.com (A.D.); mezzano.sergioa@gmail.com (S.M.); 2Renal, Vascular and Diabetes Research Laboratory, IIS-Fundación Jiménez Díaz Universidad Autónoma de Madrid, Spanish Biomedical Research Centre in Diabetes and Associated Metabolic Disorders (CIBERDEM), 28040 Madrid, Spain; susana.bu@hotmail.com (S.B.); laura.lsanz@quironsalud.es (L.L.-S.); luna.jimenez@quironsalud.es (L.J.-C.); JEgido@quironsalud.es (J.E.)

**Keywords:** diabetic nephropathy, inflammation, albuminuria, NF-κB pathway, BTBR ob/ob mice

## Abstract

Diabetic nephropathy (DN) is a multifactorial disease characterized by hyperglycemia and close interaction of hemodynamic, metabolic and inflammatory factors. Nuclear factor-κB (NF-κB) is a principal matchmaker linking hyperglycemia and inflammation. The present work investigates the cell-permeable peptide containing the inhibitor of kappa B kinase γ (IKKγ)/NF-κB essential modulator (NEMO)-binding domain (NBD) as therapeutic option to modulate inflammation in a preclinical model of type 2 diabetes (T2D) with DN. Black and tan, brachyuric obese/obese mice were randomized into 4 interventions groups: Active NBD peptide (10 and 6 µg/g body weight); Inactive mutant peptide (10 µg/g); and vehicle control. In vivo/ex vivo fluorescence imaging revealed efficient delivery of NBD peptide, systemic biodistribution and selective renal metabolization. In vivo administration of active NBD peptide improved albuminuria (>40% reduction on average) and kidney damage, decreased podocyte loss and basement membrane thickness, and modulated the expression of proinflammatory and oxidative stress markers. In vitro, NBD blocked IKK-mediated NF-κB induction and target gene expression in mesangial cells exposed to diabetic-like milieu. These results constitute the first nephroprotective effect of NBD peptide in a T2D mouse model that recapitulates the kidney lesions observed in DN patients. Targeting IKK-dependent NF-κB activation could be a therapeutic strategy to combat kidney inflammation in DN.

## 1. Introduction

Type 2 diabetes (T2D) is one of the biggest problems in global public health and considered a pandemic problem both in emerging nations and industrialized countries [1,2]. Although sodium-glucose transport protein 2 inhibitors and glucagon like-peptide receptors agonist allow better metabolic control in these patients, diabetic nephropathy (DN) remains the main cause of end-stage renal disease worldwide [3,4]. Novel anti-inflammatory therapeutic strategies could be proposed, especially in patients with rapidly progressive phenotypes such as genetic susceptibility, extreme obesity, poor metabolic control and concomitant inflammatory disease, among others [5,6].

The presence of systemic and local inflammatory milieu is a critical factor in the progression of diabetic kidney disease [6,7,8]. In T2D, chronic hyperglycemia, salt-sensitive hypertension and obesity produce metabolic, hemodynamic and lipotoxic effects described as main activators of intracellular signaling pathways such as nuclear factor kappa–B (NF-κB), Janus Kinase/Signal Transducers and Activators of Transcription (JAK/STAT) and nuclear factor erythroid 2-related factor 2/heme-oxigenase-1(Nrf2/Hmox-1) [9,10,11,12]. In particular, activation of proinflammatory NF-κB pathway has been revealed as a key molecular system involved in pathologic induction of kidney inflammation during DN [7,8].

NF-κB is an evolutionary conserved signaling pathway with an essential role in numerous biologic responses whose activation has been linked to different pathologies [13,14,15]. NF-κB is a family of proteins consisting of 15 members, exerting their action through the homo or heterodimerization of five Rel family transcription factors: p50 (or NF-κB_1_), p52 (or NF-κB_2_), p65 (or Rel-A), c-Rel and Rel-B. Several inflammatory stimuli such as hyperglycemia, advanced glycation end products, cytokines, pathogen- and danger-associated molecular patterns, CD40 ligand, and receptor activator of NF-κB ligand can activate the canonical and non-canonical NF-κB pathway [16].

NF-κB regulates the expression of distinct and overlapping subsets of genes involved in immune and inflammatory response, growth development and cellular survival [17]. Over-expression of NF-κB and proinflammatory target genes is observed in early stages of DN in both preclinical models and kidney biopsies of patients with DN [18,19,20,21,22,23]. Indeed, elevated levels of p65 correlate with upregulation of cytokines (Tumor necrosis factor (TNF)α, and Interleukin (IL)-1β) and chemokines (C–C motif chemokine ligand (CCL) 2 and, CCL5), which mediate inflammatory cell recruitment, promote local cytoskeleton arrangement and influence renal cell proliferation, apoptosis and fibrosis [18,23]. These adaptive cellular changes generate tissue remodeling, responsible for histopathologic features classically observed in DN as podocytopenia, mesangial expansion, glomerular and tubular basement membrane thickening, nodular glomerulosclerosis, tubular atrophy, inflammatory infiltrate and interstitial fibrosis [10,24].

The canonical control of NF-κB pathway is mediated by the inhibitory κB (IκB) family members (IκBα, IκBβ, IκBγ and Bcl-3) that retain NF-κB dimer in the cytoplasm as an inactive complex [17]. In this sense, the IκB kinase (IKK) complex is a key regulatory step in the NF-κB activation cascade. IKK consists of 2 catalytic subunits, IKKα (or IKK1) and IKKβ (or IKK2) and the regulatory scaffold protein IKKγ (or NF-κB essential modulator; NEMO) [25]. NF-κB pathway activation is initiated by IKK-mediated phosphorylation of IκB subunit, which induces its polyubiquitination and proteasome degradation. NF-κB transcription factors, mainly the p65–p50 heterodimer, are then released and translocated to the nucleus where they regulate gene transcription [14,25]. Studies have identified a specific region in the C-terminal of IKKα (L738–L743) and IKKβ (L737–L742) that is essential for adequate assembly of the IKKα/β–NEMO complex, accordingly named the NEMO-binding domain (NBD) [26,27]. Based on this region, several cell-permeable NBD peptides have been characterized, thus providing an opportunity to selectively abrogate proinflammatory NF-κB activity in preclinical experimental models of inflammatory diseases [27,28,29,30,31,32].

The aim of the present work was to study the potential beneficial effects of the modulation of key signaling pathways in experimental T2D with DN, focusing on the inhibition of proinflammatory NF-κB pathway. To do this, we first analyzed the association of NF-κB activation levels with renal parameters in the black and tan, brachyuric (BTBR) obese/obese (ob/ob) mouse—an obesity-associated T2D (diabesity) model that offers great translational opportunities due to its similar hallmarks of human DN [33,34]. Second, we investigated the cell-permeable NBD peptide as therapeutic option able to inhibit proinflammatory NF-κB activity and improve diabetic kidney disease in the preclinical T2D model.

## 2. Results

### 2.1. In Vivo/Ex Vivo Biodistribution of NBD Peptide in BTBR Ob/Ob Mice

Optical fluorescence imaging is a new tool for the preclinical study of pharmacological approach in small animals and human surgery intervention [35,36]. Near-infrared (NIR) fluorophore IRDye 800Cw allows high contrast, sensitivity and extinction coefficient for in vivo studies [36].

Figure 1A schematizes the generation of NBD peptide conjugated to IRDye 800Cw fluorophore and the purification by reverse phase high-performance liquid chromatography (HPLC) before image detection in Odyssey CLx^®^ system (Figure 1B,C). In addition, HPLC analysis of mouse urine samples evidenced the renal metabolization of NBD peptide–fluorophore conjugate (Figure 1D).

The administration of fluorophore-conjugated NBD peptide showed rapid systemic distribution after intravascular, subcutaneous and intraperitoneal injection (starting 15 min post-injection; Figure 2A). Ex vivo image analysis at 4 h post-injection revealed peptide incorporation in all the organs evaluated, evidencing higher uptake intensity by liver and kidney (Figure 2B). In addition, long-term pharmacokinetic behavior revealed that NBD peptide was detected for up to 48 h following intraperitoneal administration, with high accumulation in bladder indicating predominant renal elimination (Figure 2C). Based on the stability and kidney biodistribution, intraperitoneal route and three times per week dosing were chosen for intervention experiments.

### 2.2. Cell-Permeable NBD Peptide Reduced Albuminuria and Morphologic Kidney Lesions in BTBR Ob/Ob Mice

Preliminary data confirmed progressive obesity, hyperglycemia and kidney damage in BTBR ob/ob mice [33,37]. Although BTBR ob/ob is described as advanced DN model, albuminuria, podocytopenia, glomerular and tubular histopathologic changes were observed as early as 12 weeks of age. The great translational power of this model allows us to extrapolate the pathogenic mechanisms of human DN. Compared with BTBR wild type mice, kidneys from BTBR ob/ob mice showed a marked overactivation of cell signaling pathways such as NF-κB (phosphorylated-p65; *p*-p65), JAK/STAT (*p*-STAT3) and Nrf2/Hmox-1 (*p*-Nrf2 and Hmox-1) (Figure 3A–E). Furthermore, intraglomerular infiltrate of CD3+ lymphocytes and F4/80+ macrophages was observed in BTBR ob/ob mice (Figure 3F,G). Noteworthy, intraglomerular NF-κB activation revealed a positive correlation with these renal findings (Figure 3C–G).

For this reason, we further explored the therapeutic potential of cell-permeable NBD peptide targeting NF-κB in the diabesity model. Therefore, diabetic mice at 6 weeks of age were treated intraperitoneally with active NBD cell permeable peptide at two different doses (6 and 10 µg/g body weight) for 7 weeks, using vehicle (acetonitrile ≤ 0.25%) and inactive mutant peptide (Mut 10 µg/g body weight) as control groups.

No significant differences among groups were observed in body weight (Figure 4A) and glycemia (Figure 4B) during the intervention period. At the end of the follow-up period, changes in kidney weight, serum creatinine, uric acid and lipids were not significant, except for total cholesterol in NBD 10µg group (Figure 4C and Table 1). Remarkably, both doses of NBD peptide reduced albuminuria by 40–46% compared to vehicle control group (Figure 4D).

The severity of histopathologic lesions in BTBR ob/ob mouse kidney was evaluated through semiquantitative scoring. In the vehicle group, mesangial expansion and glomerulomegaly were the most relevant glomerular changes. Arteriolar hyalinosis was also found in some mice. At tubulointerstitial level, mild focal inflammatory infiltrate and tubular flattening were observed. Noteworthy, administration of active NBD peptide significantly reduced glomerular and tubulointerstitial lesions in diabetic mice (Figure 4E–H). 

### 2.3. Cell-Permeable NBD Peptide Reduced Podocyte Damage and Basement Membrane Thickness in BTBR Ob/Ob Mouse Kidney

DN is one of the main causes of podocytopathy described in nephrology, owing its current incidence in the worldwide population [38,39]. The total podocyte count was assessed by Wilms’ tumor protein-1 (WT-1) immunostaining in kidney sections. Only intraglomerular positive cells were quantified, discarding WT-1^+^ parietal cells. Compared with vehicle group, administration of cell-permeable NBD peptide significantly increased total podocyte content in a dose-dependent manner (Figure 5A). Furthermore, ultrastructural studies by transmission electron microscopy evidenced that both doses of NBD peptide improved podocyte foot process, pedicellar effacement and also regularized the outline and thickness of glomerular basement membrane (Figure 5B). Nevertheless, the thickening of tubular basement membrane, an indicator of diabetic tubulointerstitial disease, was only reduced in mice receiving a low dose of NBD peptide, with a mild irregular thickening observed in higher dose (Figure 5C).

### 2.4. Cell-Permeable NBD Peptide Modulated the Proinflammatory and Oxidative Stress Markers in BTBR Ob/Ob Mice and Cultured Cells

Real-time PCR analyses in diabetic kidneys evidenced a significant decrease in the gene expression of STAT transcription factors (*Stat1* and *Stat3*), inflammatory cytokines (*Tnfα*, *Il-1β* and *Il-12*) and chemokines (*Ccl2*, *Ccl5* and C–X–C motif chemokine ligand 10; *Cxcl10*) in NBD-treated mice compared to vehicle group (Figure 6A). Cell-permeable NBD peptide also modified redox gene expression in the kidney by reducing pro-oxidant enzyme NADPH oxidase 4 (*Nox4*) and increasing Nrf2-dependent antioxidant genes (*Nrf2/Nfe2l2* and superoxide dismutase 1; *Sod1*) (Figure 6B).

In order to confirm the in vivo findings, we performed in vitro studies using mesangial cells and macrophages under hyperglycemic and/or inflammatory conditions. In mesangial cells, NBD peptide disrupted the interaction of NEMO and IKKα/β in the kinase complex (Figure 7A) and prevented the nuclear translocation of p65 subunit (Figure 7B) induced by high-glucose in combination with TNFα. Real-time PCR revealed a dose-dependent inhibition of NF-κB-dependent genes (*Ccl2*, *Ccl5* and *Cxcl10*) by NBD peptide (Figure 7C), while the inactive Mut peptide had no significant effect. Furthermore, pretreatment with NBD reduced the NOX activity in macrophages (Figure 7D). These data suggest that the functional and structural renal improvement observed in NBD-treated mice may be due, at least in part, to the modulation of the local inflammation and oxidative stress.

## 3. Discussion

The symptomatic silence of DN only allows clinical recognition through the presence of microalbuminuria or estimated glomerular filtration rate (eGFR), main markers of kidney damage progression [40]. New research shows that certain patients with preserved eGFR do not develop proteinuria, which makes the diagnosis much more complex, and therefore proteinuria is being considered a marker of disease progression rather than a diagnostic marker [41,42,43]. For this reason, the early biomarkers validation in the DN progression is one of the most relevant challenges for the coming years [44]. At this point, the incorporation of preclinical models that represent reliably the characteristic findings of DN is a necessary pillar in the search of novel biomarkers and therapeutic targets of the disease [45]. In these sense, the BTBR ob/ob mouse model used in this study meets most of the criteria established by the Animal models diabetic complications consortium, being a valuable preclinical tool for the development of new therapeutic strategies to quickly translate the results into clinical practice [33]. This suitable preclinical model of DN recapitulating the human disease helped us to demonstrate the renoprotective potential of NF-κB-targeting peptide in T2D.

The early activation of cell signaling pathways related to inflammation and oxidative stress is directly correlated with hyperglycemia and the increase in adipose tissue dependent-body weight, and is actually described as meta-inflammation [46,47]. In this insulin-resistant and hyperglycemic context, NF-κB pathway is strongly associated with proliferative and inflammatory response in diabetic kidney, as well as cell migration, differentiation and apoptosis processes that involve both renal infiltrating and resident (mesangial, endothelial, podocyte and tubular) cells [22,48].

Previous studies in human biopsies and experimental models of DN have reported high activation levels of NF-κB family members and inflammatory target genes such as cytokines, chemokines, adhesion molecules, advanced glycation end-products and transcription factors, among others [12,18,20,29]. This is in line with an increased activity of NF-κB found in our study, both in diabetic BTBR ob/ob kidneys and in cultured cells under hyperglycemic/inflammatory conditions. We also found a positive correlation between NF-κB levels and the renal content of activated STAT3 and Nrf2 pathways and inflammatory cells. This finding prompted us to explore the inhibition of NF-κB pathway through a cell-permeable peptide derived from the IKKα/β domain as a feasible strategy to delay DN evolution in T2D.

Previous studies in T1D model by streptozotocin injection in apolipoprotein E-deficient mice have proved that NBD peptide administration ameliorates diabetic kidney damage and atherosclerosis through the modulation of systemic and local inflammation [29]. Although our results are in agreement with these findings, the clinical significance of our study is the superior protective effect of NBD peptide in a T2D mouse model that recapitulates the human pathology. Indeed, systemic administration of NBD peptide to BTBR ob/ob mice caused a significant reduction of albuminuria (>40% reduction on average), ameliorated the histopathologic glomerular and tubulointerstitial damage and also attenuated podocytopenia, pedicellar effacement and basement membrane thickening. In addition.in vivo/ex vivo pharmacokinetics and biodistribution revealed selective renal tissue uptake at both short- and long-intervention periods. Finally, mechanistic in vitro studies confirmed that NBD peptide is a potent inhibitor of IKK-dependent NF-κB activation and target gene expression in mesangial cells and macrophages under diabetic conditions.

Besides its anti-inflammatory action, NBD peptide also demonstrated antioxidant properties as evidenced by the induction of Nrf2-dependent pathway and antioxidant enzymes, as well as the suppression of pro-oxidant activity and ROS generation. Similar findings were reported in LPS-induced acute lung injury model, where NBD administration downregulated oxidative stress markers and pro-oxidant enzymes (Nox1/2/4) and improved Sod1 activity and total antioxidant capacity [49]. In a preclinical model of injury from intracerebral hemorrhage, NBD peptide relieved microglial inflammation and oxidative stress [30]. Therefore, the favorable effect of the NBD peptide is not only due to the inhibition of proinflammatory NF-κB signaling, but also through the restoration of redox balance. Our findings in the context of diabetes-induced kidney disease are consistent with other studies emphasizing the powerful therapeutic effects of different NBD-based strategies in preclinical models of Duchenne muscular dystrophy [28], Parkinson’s disease [50], arthritis [27], breast cancer [31] and ischemic acute kidney injury [51]. A recent phase I trial of NBD peptide administration in dogs with large B-cell lymphoma demonstrated safety and efficacy [32], offering hope for translation to human disease.

In conclusion, our results constitute the first nephroprotective effect of cell-permeable NBD peptide in a T2D model that recapitulates the kidney lesions observed in patients with DN, with marked reduction of albuminuria and morphologic renal lesions. Targeting selective inhibition of canonical NF-κB pathway IKK-dependent could be a therapeutic strategy to combat kidney inflammation in DN patients.

## 4. Materials and Methods

### 4.1. Synthesis of Cell-Permeable Peptides

The active NBD peptide containing a cationic cell-penetrating sequence (octalysine) fused to the IKKβ NBD region (TALDWSWLQTE) through a diglycine spacer was synthesized as described [28,29]. A biologically inactive (or less active) mutant peptide with two tryptophan-to-arginine substitutions was used. All peptides were synthesized by ProteoGenix (Schiltigheim, France). Lyophilized peptides were dissolved in acetonitrile and stock solutions were diluted in physiological saline solution before use.

For this study, male BTBR ob/ob diabetic mice were used. Breeding pairs BTBR heterozygotes (BTBR ob^+/−^) were purchased from Jackson Laboratories (Bar Harbor, ME) and housed at a density of four animals per cage in a temperature-controlled room (20–22 °C) with 12-h light–dark cycles. Ad libitum water and standard food were available. The experimental protocol was approved (25 February 2016) by the Ethics Committee for Animal Experiments of the University Austral of Chile (Permit N° 245–2016) according to National Institutes of Health guidelines.

Six-week-old male BTBR ob/ob mice were randomized into the following groups: (i) Active NBD peptide (NBD groups) at doses of 6 µg/g body weight (NBD 6 µg group; n = 8) and 10 µg/g body weight (NBD 10 µg group; n = 7); (ii) Inactive mutant peptide at dose of 10 µg/g (Mut group; n = 8); (iii) Vehicle at final acetonitrile concentration ≤0.25% in physiological saline solution (Veh group; n = 6). All groups received three intraperitoneal injections per week for seven weeks of intervention. At the end of the intervention period, all mice were analyzed and euthanized for respective analyses. In some experiments, additional for treatment groups and according to the principle of animal experimental research (the 3Rs concept), age-matched BTBR ob/ob and wild type mice kidney samples from 12-week-old (n = 5–6 mice/group) were included as a reference and immunohistochemistry analyses.

Blood glucose (Accu-Chek glucometer, Roche Diagnostics, Rotkreuz, Switzerland) and body weight levels were monitored once weekly during the follow-up period. Serum cholesterol, triglyceride and uric acid levels were measured by automated procedures. Total cholesterol, triglycerides, uric acid and serum/urine creatinine levels were measured by Jaffé reaction (LiquiColor, Wiesbaden, Germany). Albuminuria was analyzed by ELISA (Mouse albumin ALPCO, Salem, NH, USA) and corrected for urinary creatinine values to obtain urinary albumin/creatinine ratio (UACR).

### 4.2. In Vivo and Ex Vivo Optical Fluorescence Imaging

The pharmacokinetic behavior of NBD peptide was performed through non-invasive in vivo/ex vivo study with NIR fluorophore IRDye 800CW NHS ester reactive (LI-COR Biosciences, Lincoln, NE, USA). IRDye 800CW Ester Reactive binds primary amines of amino acid sequence such as lysine and due to its selective range of visualization (λexc = 680 nm y λem = 820 nm) with minimal intrinsic interference (autofluorescence). Purification of NBD peptide and IRDye 800CW-conjugated NBD peptide was performed through reverse phase HPLC UltiMate 3000RS system, coupled to a TSQ Vantage mass spectrometer (Thermo Fisher Scientific, Waltham, MA, USA). Prior to animal injection (intraperitoneal, ventral subcutaneous and intravascular), IRDye 800CW-conjugated NBD peptide was evaporated to dryness in SpeedVac^®^ for 2 h, quantified by Pierce BCA Protein Assay Kit (Thermo Fisher ScientificWaltham, MA, USA) and dissolved in PBS 1× at a final concentration of 6 µg/g body weight.

BTBR ob/ob mice (12 weeks-old, n = 3) were anesthetized with oxygen/isoflurane mixture administered by SmartFlow^®^ vaporizer unit in induction chamber and then incorporated in a built-in heating base at 33 °C in Odyssey CLx infrared imaging system (LI-COR Biosciences, Lincoln, NE, USA) for images detection. After 4 h post-injection, BTBR ob/ob mice were euthanized; heart, liver, inguinal visceral fat tissue, eyeball and kidney were collected for *ex vivo* imaging kidney and heart were cut along the coronal plane.

### 4.3. Histological Analysis and Immunohistochemistry

For euthanasia, mice were anesthetized with 2% 2,2,2-tribromoethanol (Sigma-Aldrich, Burlington, MA, USA) dissolved in 2-methyl-2-butanol (Sigma-Aldrich, Burlington, MA, USA). Blood samples were taken for serum collection and both kidneys were removed, decapsulated and hemisected along their perihilar sagittal plane. One-half of each kidney (right and left) was fixed in 4% formaldehyde. A small portion of the renal cortex was embedded in 2% glutaraldehyde for transmission electron microscopy. The remaining portion was stored immediately in liquid N_2_ and processed for RNA extraction.

The kidneys were fixed in 4% formaldehyde, embedded in paraffin and cut in 3–4 μm tissue sections for histology (Periodic acid Schiff (PAS) and Masson’s trichrome staining) and immunohistochemistry studies. Renal damage was examined in a blinded manner and scored using a semiquantitative histopathologic scale (0 to 4) to evaluate glomerular and tubulointerstitial lesions as previously described [52]. The primary antibodies for immunodetection studies were: WT-1 podocyte marker (Agilent Cat# M3561, RRID:AB_2304486, dilution 1:100, Santa Clara, CA, USA), F4/80 monocytes/macrophages (Bio-Rad Cat# MCA497, RRID:AB_2098196, dilution 1:70, Hercules, CA, USA), CD3 T lymphocytes (Agilent Cat# M7254, RRID:AB_2631163, dilution 1:100Santa Clara, CA, USA), *p*-p65 serine 536 (Santa Cruz Biotechnology Cat# sc-33020, RRID:AB_2179018, dilution 1:100, Santa Cruz, CA, USA), *p*-STAT3 tyrosine 705 (Cell Signaling Technology Cat# 9134, RRID:AB_331589, dilution 1:100, Leiden, The Netherlands) and *p*-Nrf2 serine 40 (Abcam Cat# ab76026, RRID:AB_1524049, dilution 1:2000, Cambridge, UK). For WT-1 and *p*-Nrf2, incubation for primary antibody was carried out using the M.O.M. immunodetection kit (Vector Laboratories Cat# BMK-2202, RRID:AB_2336833, Peterborough, UK) and Vectastain Elite ABC HRP kit R.T.U. (Vector Laboratories Cat# PK-7100, RRID:AB_2336827). All other primary antibodies were detected by indirect immunoperoxidase. Sections were revealed with ImmPACT DAB peroxidase substrate (Vector Laboratories Cat# SK-4105, RRID:AB_2336520), counterstained with Harris hematoxylin and late dehydration for subsequent mounting in non-aqueous medium Canada Balsam (Cell Marque Rocklin, CA, USA). The WT-1 positive cells were quantified using Image Pro-Plus software (Media Cybernetics, Gainesville, FL, USA https://www.mediacy.com/imageproplus) and expressed as number of positive cells per glomerular cross section.

For analysis by transmission electron microscopy, the kidney tissue was fixed in 2% glutaraldehyde (Merck, Darmstad, Germany), post-fixed with 1% osmium tetroxide (Ted Pella, Inc., Redding, CA, USA) and observed under a Philips Tecnai 12 electron microscope (Philips, Eindhoven, The Netherlands) operated at 80 kV.

### 4.4. Cell Cultures

Mouse mesangial cell line (SV40 MES13; ATCC Cat# CRL-1927, RRID:CVCL_5368) was maintained in a 3:1 mixture of DMEM and Ham’s F12 medium containing 14-mM HEPES, 100 U/mL penicillin, 100 μg/mL streptomycin and 2-mM L-glutamine and 5% Fetal Bovine Serum (FBS). Mouse macrophage cell line (RAW 264.7; ATCC Cat# TIB-71, RRID:CVCL_0493) was maintained in DMEM with 10% FBS. Cells were made quiescent by 24 h incubation in medium without FBS and then treated for 90 min with NBD and Mut peptides (25–50 µM) before stimulation with 30-mM D-glucose plus 100 U/mL TNFα (HG + TNFα) or 1-μg/mL LPS.

### 4.5. Western Blot and Immunoprecipitation

Nuclear and total protein extracts from mesangial cells were prepared as described elsewhere [29]. Total proteins (200 μg) were incubated with 2 μg NEMO antibody (BD Biosciences Cat# 557383, RRID:AB_396669, Oxford, UK) overnight at 4 °C and then with 20 μL of protein A/G-PLUS-Agarose (Santa Cruz Biotechnology Cat# sc-2003, RRID:AB_10201400Santa Cruz, CA, USA) for 2 h at 4 °C. After washing, immunoprecipitates were eluted and developed by polyacrylamide gel electrophoresis, then transferred onto polyvinylidene difluoride membranes and immunoblotted with IKKα/β (Santa Cruz Biotechnology Cat# sc-7607, RRID:AB_675667, Santa Cruz, CA, USA) and NEMO antibodies. Nuclear fractions (30 μg) were immunoblotted for p65 (Santa Cruz Biotechnology Cat# sc-372, RRID:AB_632037, Santa Cruz, CA, USA) and histone H3 (Cell Signaling Technology Cat# 5192, RRID:AB_1950410, Leiden, The Netherlands) antibodies. Bands were visualized by enhanced chemiluminescence system, quantified (Quantity One, Bio-Rad, Hercules, CA, USA) and normalized by loading controls.

### 4.6. NOX Activity Assay

NOX-dependent superoxide anion generation in macrophages was measured by the lucigenin chemiluminescence assay as described [53]. Briefly, cells were homogenized in 50-mM phosphate buffer containing 0.01-mM EDTA, 0.32-mM sucrose and 0.1% protease inhibitor cocktail. Lysates were transferred to Röhren tubes containing 5-μM lucigenin and 100-μM NADPH (Sigma-Aldrich, Burlington, MA, USA) and chemiluminescence was measured with a luminometer (Berthold Technologies, Bad Wildbad, Germany) by counting the photon emission at 5-s intervals over 5 min. Values were expressed as relative light units (RLU) per mg of protein.

### 4.7. mRNA Expression

Total RNA from kidney tissue and cells was isolated with TRIzol reagent (Ambion Inc, New Haven, CT, USA). cDNA was synthesized by ImProm-II TM Reverse Transcription System (Promega, Madison, Wisconsin, USA) using 2 μg of total RNA primed with random hexamer primers. The analysis of quantitative gene expression was performed on a Rotor-Gene Q (Qiagen, Venlo, The Netherlands) using primers designed by Integrated DNA Technologies (IDT, Coralville, IA, USA) and KAPA SYBR FAST Universal Kit (Kapa Biosystems, Wilmington, MA, USA). The expression of target genes was analyzed in duplicate and normalized by 18S rRNA housekeeping gene. The primer sequences used in this study are detailed in Table 2.

### 4.8. Statistical Analysis

The data were expressed as scatter dot plots of total number of experiments. Statistical analyses were performed using the nonparametric Mann–Whitney U test for comparison between two groups, one-way ANOVA with Dunnett’s posthoc test for multiple comparisons and Pearson coefficient test for correlation studies, considering differences statistically significant those with values of *p*<0.05. All graphs and statistical tests were performed using GraphPad Prism 6 software.

## Figures and Tables

**Figure 1 ijms-21-04225-f001:**
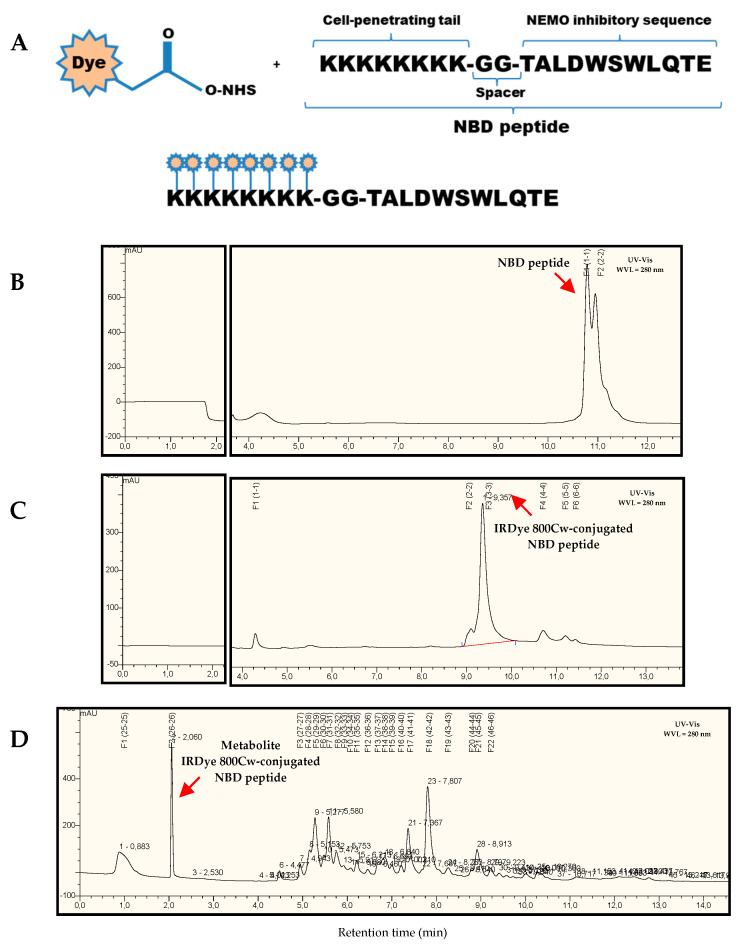
Sequence and characterization of IRDye 800Cw-conjugated IκB Kinaseγ/NF-κB essential modulator (IKKγ/NEMO)-binding domain (NBD) peptide.(**A**) Schematic representation of the labeling reaction of the fluorophore dye (IRDye 800Cw NHS Ester) with primary amine groups (Lys) of NBD peptide. Representative high-performance liquid chromatography (HPLC) chromatograms of NBD peptide and (**B**) IRDye 800Cw-conjugated NBD peptide (**C**) in PBS; (**D**) Metabolite profile from mouse urine sample at 4 h after intraperitoneal injection.

**Figure 2 ijms-21-04225-f002:**
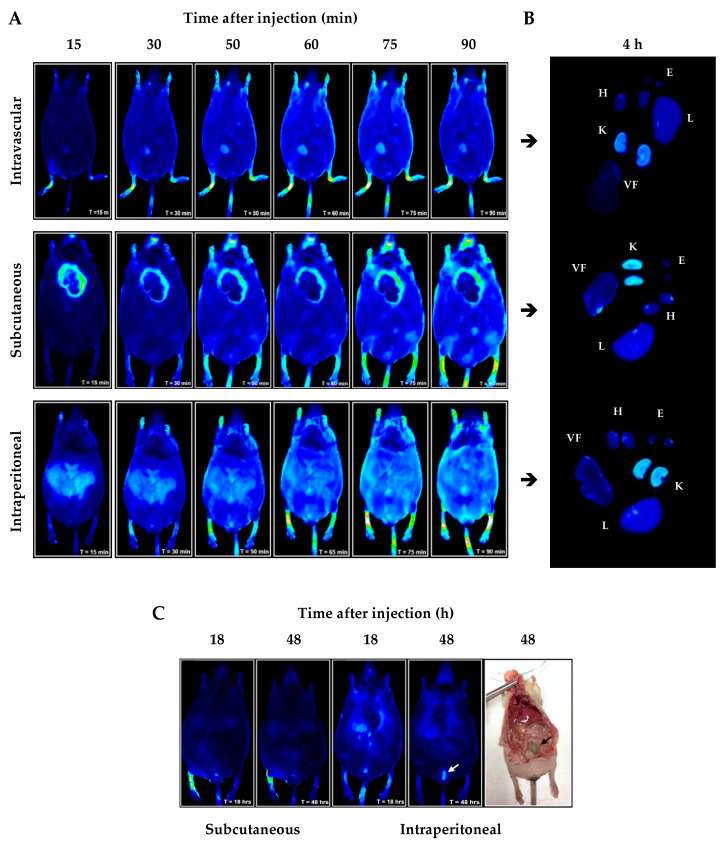
*In vivo/ex vivo* biodistribution of fluorochrome-conjugated NBD peptide in black and tan brachyuric (BTBR) ob/ob mice. (**A**) Short-time evaluation of the in vivo biodistribution by three different administration routes; (**B**) four hours post-injection, ex vivo analysis demonstrates selective kidney metabolization; (**C**) long-time evaluation of the in vivo biodistribution confirms the presence of fluorophore-conjugated NBD peptide after 48 h of administration. Arrows indicate bladder (white) and fecal (black) excretion. Abbreviations: H—heart; K—kidney; L—liver; E—eye and VF—visceral fat.

**Figure 3 ijms-21-04225-f003:**
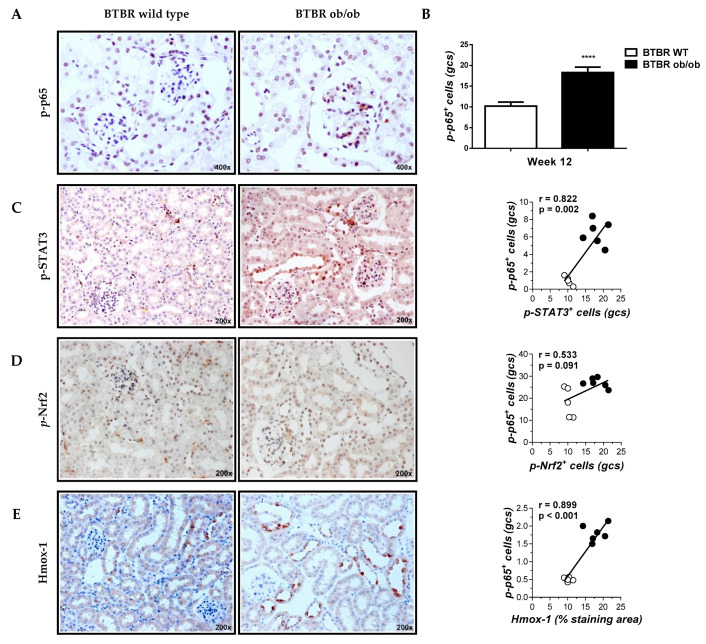
Correlation between glomerular Nuclear factor-κB (NF-κB) activation and immunohistochemical kidney markers related to inflammation. Representative images (magnification 400× and 200×) of immunodetection of *p*-p65 (**A**), *p*-STAT3 (**B**), *p*-Nrf2 (**C**), Hmo×-1 (**D**), CD3 T cells (**F**) and F4/80 macrophages (**G**) in kidney sections; (**B**) Quantification of *p*-p65 positive cells in glomerular cross sections (gcs). Data shown as mean ± SD of n = 5–6 mice/group. **** *p* < 0.001 vs. BTBR wild type (WT) mice; (**C**–**G**) Pearson’s correlation analysis of glomerular *p*-p65 with different histological markers. White symbols indicate BTBR wild type; black symbols, BTBR ob/ob.

**Figure 4 ijms-21-04225-f004:**
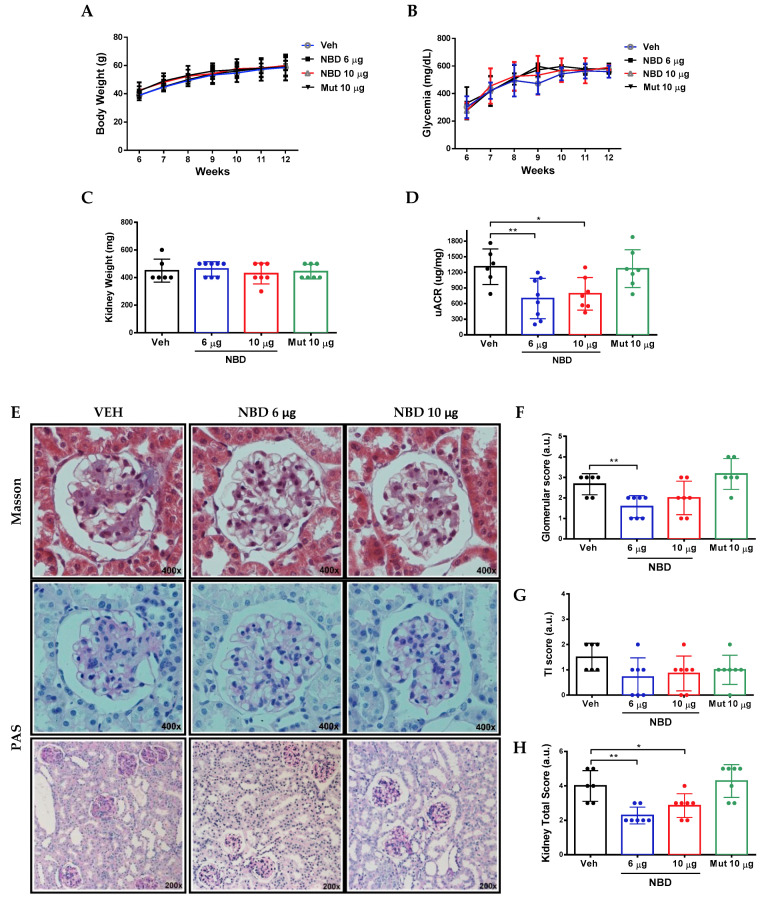
Cell-permeable NBD peptide reduced albuminuria and morphologic kidney lesions in BTBR ob/ob mice. Evolution of (**A**) body weight and (**B**) glycemia in treated groups. At the end of the study, (**C**) kidney weight, (**D**) urinary albumin creatinine ratio and (**E**) histological changes were evaluated. Semiquantitative assessment of (**F**) glomerular, (**G**) tubulointerstitial and (**H**) total kidney score (Magnification 200× and 400×. Data shown as scatter dot plots and mean ± SD of each group (n = 6–8 mice/group). * *p* < 0.05; ** *p* < 0.01 vs. diabetic vehicle control.

**Figure 5 ijms-21-04225-f005:**
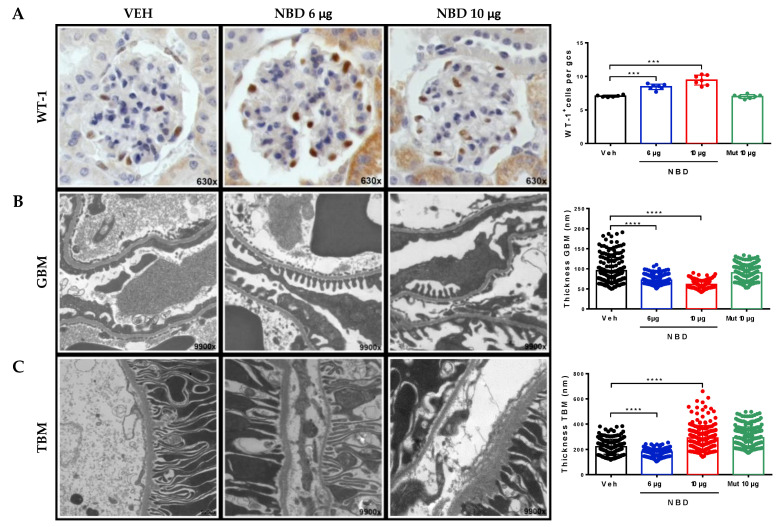
Cell-permeable NBD peptide decreased podocyte loss and basement membrane thickness in BTBR ob/ob mouse kidney. (**A**) Representative images of WT-1 immunohistochemistry (magnification 630×) and quantification of WT-1^+^ cells in glomerular cross section (gcs). Transmission electron microscopy images (magnification 9900×) and measurement of (**B**) glomerular basement membrane (GBM) and (**C**) tubular basement membrane (TBM) thickness. Data shown as scatter dot plots and mean ± SD of each group (n = 6–8 mice/group). *** *p* < 0.001; **** *p* < 0.0001 vs. diabetic vehicle control.

**Figure 6 ijms-21-04225-f006:**
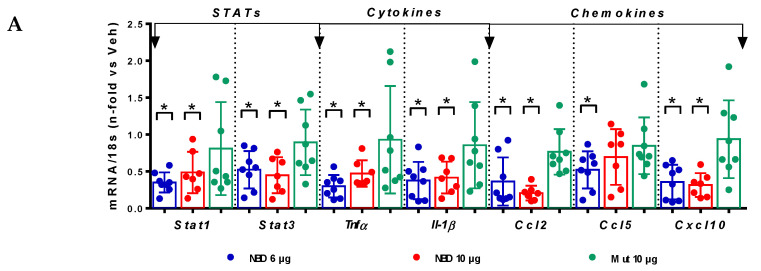
Cell-permeable NBD peptide modulated the renal gene expression of inflammatory and oxidative stress markers in BTBR ob/ob mice. (**A**) Real-time PCR analysis of STAT-dependent and (**B**) redox balance genes, in kidney samples. Values normalized in each sample by endogenous control gene 18s and expressed as n-fold of the average value from diabetic vehicle control. Data shown as scatter dot plots and mean ± SD of each group (n = 6–8 mice/group). *—*p* <0.05; **—*p* < 0.01; ***—*p* < 0.001 vs. diabetic vehicle control.

**Figure 7 ijms-21-04225-f007:**
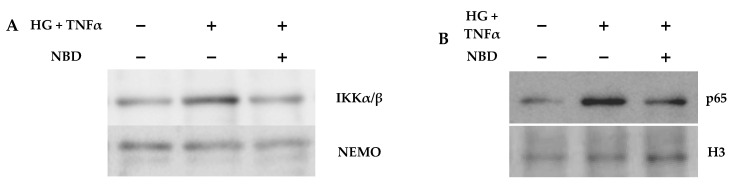
Cell-permeable NBD peptide inhibits NF-κB activation, inflammation and oxidative stress in vitro. (**A**) Immunoprecipitation of NEMO with IKKα/β subunits in mesangial cells treated with NBD peptide (50 µM, 90 min) before stimulation with high-glucose and TNFα (HGG+ TNFα, 2 h); (**B**) western blot analysis of nuclear p65 protein (loading control, histone H3) in mesangial cells; (**C**) real-time PCR analysis of proinflammatory genes (normalized to 18S) in mesangial cells pretreated with NBD and Mut peptides prior to stimulation (HG + TNFα, 24 h); (**D**) measurement of NADPH oxidase activity (expressed as relative light units (RLU) per mg) in macrophage cell lysates after 3 h of stimulation with lipopolysaccharide (LPS) in the presence or absence of NBD and Mut peptides. Results presented as individual data points and mean ± SD of 4 independent experiments. *—*p* < 0.05 vs. basal; #—*p* < 0.05 vs. stimulus; $—*p* < 0.05 vs. Mut.

**Table 1 ijms-21-04225-t001:** Serum metabolic parameters of BTBR ob/ob mice at the end of the study.

Group (n)	Creatinine (mg/dL)	Total Cholesterol (mg/dL)	Triglycerides (mg/dL)	Uric Acid (mg/dL)
**Vehicle (6)**	0.80 ± 0.08	158.0 ± 9.7	214.6 ± 69.1	3.90 ± 0.36
**NBD 6 μg (8)**	0.82 ± 0.08	177.4 ± 9.8	284.6 ± 35.4	3.30 ± 0.50
**NBD 10 μg (7)**	0.86 ± 0.04	207.9 ± 10.9 *	289.7 ± 34.1	4.04 ± 0.29
**Mut 10 μg (7)**	0.74 ± 0.06	165.3 ± 10.7	239.1± 27.0	3.17 ± 0.22

Data expressed as mean ± SD. * *p* < 0.05 vs. diabetic vehicle control.

**Table 2 ijms-21-04225-t002:** Primers used for real-time PCR analysis.

Gene	5′-3′ Forward	5′-3′ Reverse
*Stat1*	TGAGATGTCCCGGATAGTGG	CGCCAGAGAGAAATTCGTGT
*Stat3*	GTCTGCAGAGTTCAAGCACCT	TCCTCAGTCACGATCAAGGAG
*Tnfα*	ATGGCCTCCCTCTCATCAG	TTTGCTACGACGTGGGCTAC
*Il-1β*	GCTGAAAGCTCTCCACCTCA	CTTGGGATCCACACTCTCCAG
*Ccl2*	AGCTCTCTCTTCCTCCACCA	GGCGTTAACTGCATCTGGCT
*Ccl5*	AGAGGACTCTGAGACAGCACA	CGAGCCATATGGTGAGGCAG
*Cxcl10*	ACTCCCCTTTACCCAGTGGA	CCACTTGAGCGAGGACTCAG
*Nfe2l2*	GATCCGCCAGCTACTCCCAGGTTG	CAGGGCAAGCGACTCATGGTCATC
*Nox4*	CCCTCCTGGCTGCATTAGTC	AACCCTCGAGGCAAAGATCC
*Sod1*	GGAACCATCCACTTCGAGCA	CTGCACTGGTACAGCCTTGT
*Catalase*	GGTGCCCCCAACTATTACCC	GAATGTCCGCACCTGAGTGA
*18s*	CCGTCGTAGTTCCGACCATAA	CAGCTTTGCAACCATACTCCC

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
