# Peer review of "Targeting NF-κB by the Cell-Permeable NEMO-Binding Domain Peptide Improves Albuminuria and Renal Lesions in an Experimental Model of Type 2 Diabetic Nephropathy"

_ijms, 2020, doi:10.3390/ijms21124225_

Round 1

Reviewer 1 Report

This is an elegantly designed and well-executed study applying a novel therapeutic intervention targeting glomerular injury in a relevant model of type 2 diabetes. I have a few comments.

  1. Why did you choose the intraperitoneal route of administration? This should be explained.
  2. What do you want the reader to observe in Figure 1, panel D: “and urine sample after 4 hours intraperitoneal injection (D).”? Indicate the relevant peak(s) with an arrow and explain this in the legend.
  3. How do you interpret the discontinuous thickening of tubular basement membrane observed in the high-dose NBD group (Figure 5A). This needs to be discussed.
  4. In mice plasma urea is a more sensitive indicator of renal function than plasma creatinine because laboratory mice have a high protein intake but a very low muscle mass. Please include urea in Table 1.
  5. In the statistical analysis I would use a Dunnett test for post-hoc analysis because all comparisons are vs. the vehicle group. This is more suitable (and less stringent) than the Bonferroni test.

Minor comments

  1. The terms “On the other hand,” (introduction, last paragraph) and “For the other hand,” (discussion, 2nd paragraph) are used inappropriately. You always need two hands when using these phrases (On the one hand, … on the other hand, …).
  2. What do the arrows indicate in Figure 2, panel C? Presumably the bladder. Please explain this in the legend.
  3. In Table 1 please indicate the units for each variable. N/group should be indicated horizontally under the units.
  4. In the discussion please indicate that in cited ref. #29 STZ induces type 1 diabetes
  5. The term in the discussion “has suffered an explosive increase” is inappropriate because you clearly regard these developments as beneficial, not detrimental. In general this section could be down-tuned a bit. It now has a style that is more appropriate for a review.
  6. In section 4.3 please include the methods used to measure uric acid, cholesterol and triglycerides.
  7. #26: delete (80-. ).
  8. #29: G-.G.C. should read Gomez-Guerrero C.
  9. #33: insert J Am Soc Nephrol.

Author Response

Reviewer 1

This is an elegantly designed and well-executed study applying a novel therapeutic intervention targeting glomerular injury in a relevant model of type 2 diabetes. I have a few comments.

Author reply: We thank the Reviewer for thorough evaluation of our manuscript and for his/her critical comments and constructive suggestions.

Why did you choose the intraperitoneal route of administration? This should be explained.

Author reply: Our short-term in vivo biodistribution studies reveal excellent pharmacokinetic behavior through subcutaneous and intraperitoneal administration routes. However, based on ex vivo and long-term studies (48 hours), we showed better systemic and renal uptake, latency, and metabolism following the intraperitoneal route of administration. We have added new lines in the text (Section 2.1) and modified the Figure 2 legend to improve the understanding of this topic in the manuscript.

What do you want the reader to observe in Figure 1, panel D: “and urine sample after 4 hours intraperitoneal injection (D).”? Indicate the relevant peak(s) with an arrow and explain this in the legend.

Author reply: In the chromatogram of the urine sample, an intensity peak with a similar retention time (9.357 min) was not detected compared to the chromatogram of the NBD-fluorophore peptide conjugate, likely due to peptide metabolization. Accordingly, two metabolites of similar intensity were detected, but with earlier retention times. Following your suggestion, we have modified the text (Section 2.1) with the aim to clarify the results of renal metabolization, and also added arrows in the chromatograms (NBD peptide in PBS 1x, fluorophore-NBD peptide conjugate and urinary profile after intraperitoneal administration) of Figure 1.

How do you interpret the discontinuous thickening of tubular basement membrane observed in the high-dose NBD group (Figure 5A). This needs to be discussed.

Author reply: The irregularity in the basement membrane thickness has been well described at the glomerular level as the shear stress and mechanical forces generated by hyperfiltration and podocyte effacement (PMID 30947190; 30334635) or degradation imbalance of extracellular matrix proteins, either due to the increase in matrix metalloproteinases or decrease of their inhibitors (tissue inhibitors of metalloproteinases) (PMID 18418356; 27582102; 16543722). However, at the tubular level, there is limited information on which could be responsible for this discontinuity. The authors believe that although it is an important event (for this reason is included in the results), it was only appreciated at high doses of NBD treatment. Therefore, taking into account the above, we have decided to improve the understanding of these results in the manuscript (Section 2.3), clarifying some relevant aspects, as suggested by the Reviewer.

In mice plasma urea is a more sensitive indicator of renal function than plasma creatinine because laboratory mice have a high protein intake but a very low muscle mass. Please include urea in Table 1.

Author reply: We agree with the Reviewer that plasma urea concentration is a reliable and sensitive indicator of renal dysfunction in mouse models. Unfortunately, those urine and serum samples are not currently available in our lab to measure urea levels.

In the statistical analysis I would use a Dunnett test for post-hoc analysis because all comparisons are vs. the vehicle group. This is more suitable (and less stringent) than the Bonferroni test.

Author reply: Following your suggestion, new statistical analysis has been performed using the post-hoc Dunnett's test for multiple comparisons.

Minor comments

The terms “On the other hand,” (introduction, last paragraph) and “For the other hand,” (discussion, 2nd paragraph) are used inappropriately. You always need two hands when using these phrases (On the one hand, … on the other hand, …).

Author reply:  Thanks for your comment. In the Introduction and Discussion sections several sentences have been reworded accordingly.

What do the arrows indicate in Figure 2, panel C? Presumably the bladder. Please explain this in the legend.

Author reply: The meaning of arrows is now explained in the Figure 2 legend.

In Table 1 please indicate the units for each variable. N/group should be indicated horizontally under the units.

Author reply: Table 1 has been modified as suggested by the Reviewer.

In the discussion please indicate that in cited ref. #29 STZ induces type 1 diabetes. The term in the discussion “has suffered an explosive increase” is inappropriate because you clearly regard these developments as beneficial, not detrimental. In general this section could be down-tuned a bit. It now has a style that is more appropriate for a review.

Author reply: Thanks for your comment. In the new version, we reorient the discussion around the findings of NBD peptides in different experimental conditions. The sentence regarding ref #29 is now completed.

In section 4.3 please include the methods used to measure uric acid, cholesterol and triglycerides.

Author reply: Done

#26: delete (80-. ).                

Author reply: Done

#29: G-.G.C. should read Gomez-Guerrero C.

Author reply: Done

#33: insert J Am Soc Nephrol.

Author reply: Done

Reviewer 2 Report

The authors investigated the cell-permeable peptide containing the IkB Kinaseγ/NF-κB essential modulator 20 -binding domain as therapeutic option to modulate inflammation in an experimental model of type 2 diabetic nephropathy. However, there are some problems had to be detail clarified 

  1. In introduction, author mention that in T2DN, chronic hyperglycemia, salt-sensitive hypertension and obesity produce metabolic, hemodynamic and lipotoxic effects described as main activators of intracellular signaling pathways such as Nuclear Factor kappa-B (NF-κB), Janus Kinase Signal Transducers and Activators of Transcription (JAK/STAT) and Nuclear Factor Erythroid 2-related Factor 2/Heme-Oxigenase-1(NRF2/HO-1). Why did author only focus on NF-κB? This pathway is reported by many previous studies and well-understand by readers.
  2. Extensive editing of English language is necessary. The article section had some problem.
  3. There was no formal hypothesis enunciated prior to the study, especially in mechanism.
  4. Figure 1 legend had some mistakes.
  5. It is necessary to report also all the new studies of the literature on this topic which are now missing. 
  6. The mechanism of pathophysiological links between Cell-permeable NBD peptide modulated the renal gene expression of inflammatory and stress oxidative is reported in some previous studies. We suggested authors to report current results in article.

Author Response

Reviewer 2

The authors investigated the cell-permeable peptide containing the IkB Kinaseγ/NF-κB essential modulator 20 -binding domain as therapeutic option to modulate inflammation in an experimental model of type 2 diabetic nephropathy. However, there are some problems had to be detail clarified

Author reply: Thank you very much for taking time in reading our work. We sincerely appreciate your valuable suggestions, which helped us to improve the quality of the manuscript.

In introduction, author mention that in T2DN, chronic hyperglycemia, salt-sensitive hypertension and obesity produce metabolic, hemodynamic and lipotoxic effects described as main activators of intracellular signaling pathways such as Nuclear Factor kappa-B (NF-κB), Janus Kinase Signal Transducers and Activators of Transcription (JAK/STAT) and Nuclear Factor Erythroid 2-related Factor 2/Heme-Oxigenase-1(NRF2/HO-1). Why did author only focus on NF-κB? This pathway is reported by many previous studies and well-understand by readers.

Author reply: For many years, our research group has focused on the activation of signaling pathways in the progression of diabetic nephropathy (DN) and others diabetic complications, both in preclinical models and human samples. We have recently published an updated review of new inflammatory pathogenic mechanisms in DN progression in this special issue (Int. J. Mol. Sci. 2020, 21(11), 3798; https://doi.org/10.3390/ijms21113798). Particularly NF-kB pathway is a key molecular system in the maintenance of the inflammatory state in DN. Modulation of the NF-kB pathway through this therapeutic approach has been evaluated by other authors in various inflammatory-based pathologies, including our early study in type 1 diabetes. Nevertheless, to our knowledge, this is the first study addressing NBD peptide effectiveness in a type 2 diabetes mouse model that recapitulates human diabetic nephropathy.

We have made changes in Introduction (second and last paragraphs) and Discussion (third and fourth paragraphs) in order to better clarify the basis of our study.

Extensive editing of English language is necessary. The article section had some problem.

Author reply: The manuscript has been revised for typo and grammatical errors.

There was no formal hypothesis enunciated prior to the study, especially in mechanism.

Author reply: Following your suggestion, we have modified the last paragraph in Introduction to clearly state the aims and working hypothesis of our study.

Figure 1 legend had some mistakes.

Author reply: We corrected Figure 1 legend.

It is necessary to report also all the new studies of the literature on this topic which are now missing.

The mechanism of pathophysiological links between Cell-permeable NBD peptide modulated the renal gene expression of inflammatory and stress oxidative is reported in some previous studies. We suggested authors to report current results in article.

Author reply: Following your suggestion, in the Discussion section we have mentioned other studies reporting the anti-inflammatory and antioxidant effect of NBD peptide, some of them at similar doses to those used in our experimental design. Please, see fifth paragraph and references #30, #49 and #51